# Assessing the Occurrence of Host-Specific Faecal Indicator Markers in Water Systems as a Function of Water, Sanitation and Hygiene Practices: A Case Study in Rural Communities of Vhembe District Municipality, South Africa

**DOI:** 10.3390/pathogens13010016

**Published:** 2023-12-23

**Authors:** Dikeledi Prudence Mothiba, Colette Mmapenya Khabo-Mmekoa, Renay Ngobeni-Nyambi, Maggy Ndombo Benteke Momba

**Affiliations:** 1Department of Environmental, Water and Earth Sciences, Tshwane University of Technology, Arcadia Campus, Private Bag X680, Pretoria 0001, South Africa; mothibadeekay@hotmail.com (D.P.M.); renayngobeni@gmail.com (R.N.-N.); 2Department of Biomedical Technology, Tshwane University of Technology, Arcadia Campus, Pretoria 0001, South Africa; mmekoakcm@tut.ac.za; 3Department of Microbiology, Stellenbosch University, Private Bag X1, Matieland, Stellenbosch 7602, South Africa

**Keywords:** faecal contamination, microbial source tracking, rural areas, waterborne diseases, water and sanitation, zoonosis

## Abstract

In settings where humans and animals closely coexist, the introduction of faecal material into unprotected water sources significantly increases the risk of contracting diarrhoeal and zoonotic waterborne diseases. The data were gathered from a survey conducted through interviews at randomly sampled villages; additionally, water samples were collected in randomly selected households and their associated feeder catchments. Molecular techniques were used, specifically qPCR, to run host-specific *Bacteroides* microbial source tracking (MST) assays for human, cattle, pig, chicken and dog faecal contamination. Unexpectedly, the qPCR assays revealed dogs to be the most prevalent (40.65%) depositor of faecal matter in unprotected surface water, followed by humans (40.63%); this finding was contradictory to survey findings indicating cattle as the leading source. At the household level, dogs (16.67%) and chickens (15.28%) played prominent roles, as was expected. Reflecting on some of the basic daily practices in households, nearly 89.00% of the population was found to store water due to erratic supply, in contrast to 93.23% using an improved water source. Additionally, a significant association was found between water, sanitation and hygiene (WASH) variables and the occurrence of MST markers after performing a bivariate linear regression. However, the inconsistency between the MST results and household surveys suggests pervasive sanitation issues, even in households without domesticated animals.

## 1. Introduction

There are many ways in which people become exposed to water that is contaminated with faecal matter, bearing disease-causing microbes. To detect this potential risk, host-associated markers have been used to identify faecal pollution in water sources [1] and in drinking water [2], with significant gains. *Bacteroides* spp. exhibit a strong host specificity and are therefore a reliable way of detecting faecal contamination in water bodies [3,4]. Their association with diarrhoea-causing pathogens in both humans and animals make them a reliable indicator of more serious source-specific and zoonotic enteric pathogens.

A study of the wildlife–livestock interface that took place in the Vhembe district revealed that transboundary disease transfer and livestock depredation by wild animals have been happening in the area and its neighbouring countries such as Zimbabwe and Mozambique and particularly in the South African Kruger National Park (KNP) [5]. Some wildlife of the KNP, including buffaloes, were found to be permanently infected with some diseases such as ticks, bovine tuberculosis and foot-and-mouth, any of which may be easily transmissible to the domesticated cow, other domestic animals and humans, should an escape happen [6]. This is particularly alarming given the geographic morphology of water sources running through the villages before emptying into the Luvuvhu River and eventually the Limpopo River, along the KNP.

Additionally, domestic animals pose health risks for their owners in rural environments. The efforts to cut back on costs, while ensuring security for their stock forces farmers to keep animals in close proximity, with very close contact daily [7]. Domesticated animals that are fed household wastes or neglected to become scavengers may increase the risk of zoonosis in a household. Ironically, in these rural areas, clinics and laboratories for animals are non-existent [8]. Small-scale farmers make up a significant portion of rural communities. There is a huge reliance on the already stressed freshwater sources for domestic use, including irrigation and husbandry [9]. It is important for the crops that are cultivated to be of good quality, as this could be a pathway for faecal–oral pathogens. Subsistence farming is often undertaken in schools as part of feeding schemes, raising funds through sales or even engaging learners in agricultural education as part of their curriculum. Previous studies have found that younger children in primary schools are often carriers of enteric parasites [10]. The chain from contaminated soil, fomites, hands, pets and vectors needs to be broken. This is especially true for households with members as young as the age of five years and below. Death by diarrhoea is a global concern in this age range, and an elevated risk exists [11]. Penakalapati et al. [12] share findings gathered through their review that flies have been associated with trachoma and poor hygiene along with animal ownership.

Habitual behaviours remain a great risk to public health [13]. It has been argued that having access to clean water and improved sanitation facilities does not necessarily equate to a diarrhoeal-disease-free community [14,15]. Basic habits such as the regular washing of hands can significantly reduce WASH-related conditions [16]. This is true when soap is used. Paradoxically, the basis remains that clean water and soap need to be available to achieve that end. Behaviour can either be looked at as the more cost-effective intervention for recognisable improvements, or as one of the hardest aspects to change when deeply rooted within a population [17], rendering it even more costly to uproot. There is a constant shift in migration from rural to urban settings or the urbanisation of existing homelands for benefits offered by the state. This inevitable pursuit of better living standards leads to an increased demand for service delivery, which the treatment plants are not able to meet. Wastewater treatment plants (WWTPs) become overloaded due to receiving elevated volumes of wastewater beyond the original design capacity.

With a large percentage of household members in rural areas earning less than ZAR 500 (USD ±27) per month [18], the capacity to afford cleaning agents is low. Many households run without a consistent or fixed basic income. Members scramble to make ends meet, with the situation aggravated in child-headed households. If there is not enough money for meals to get by, spending on sanitation products is not a reality. This includes cleaning agents and handwashing soap, with a direct impact on hygiene. Studies have shown that treating water at the point of use (POU) greatly reduced the abundance of faecal indicator bacteria and diarrhoea-causing pathogens in a village in Limpopo [19,20,21]. In another study conducted in KwaZulu-Natal, Khabo-Mmekoa et al. [22] confirmed the contamination of water that was consumed at household taps with enteric pathogenic microorganisms; the extent of the contamination was less than that observed in household container-stored water of rural dwellers, but contaminated nonetheless.

When the impact of sanitation intervention on pathogen manifestation in a household was investigated [23], no significant difference was observed between the control arm and the sanitation (intervention) arm with regard to faecal contamination. This study had introduced some hygiene improvements in rural household compounds but found minimal impact. The need for a wider coverage was established to be one of the reasons, because pathogens are transported universally. This emphasises the gravity of breaking habitual behaviour. It was reported that densely populated communities experience more frequent water contamination issues [24]. In addition to this is the lack of safe handling of wastes and excreta and, lastly, the occurrence of intensive farming.

Previous investigations in South Africa have underscored the critical issue of non-compliance of small-scale WWTPs and drinking water treatment plants (DWTPs) in Limpopo Province posing a threat to public health [25,26]. The deficiency was notable when the compliance with microbiological and chemical parameters was not met; even more so when pertinent data could not be managed or projected. This lapse shed light on the quality of operational performance of such plants. In 2016, Limpopo Province recorded the highest lack of safe and reliable water supply [27]. While there has been progress over subsequent years, a substantial effort is still necessary to meet the Sustainable Development Goals (SDGs) by 2030 [28]. Active and competent authorities are essential for the success of the goal to reach full coverage of water and sanitation in South Africa.

The aim of this study was to assess the occurrence of host-specific faecal indicator markers in water systems in order to pinpoint the most prevalent contributor to faecal contamination from the catchment area to the household level in some rural areas in the Vhembe District Municipality, Limpopo Province, South Africa. We sought to answer the following questions: Is the presence of *Bacteroides* in water samples reflected in household practices? Is faecal contamination of water introduced where animals are found? Does animal ownership have an effect on the contamination of water? Are natural water sources more contaminated by domestic animals or by humans?

## 2. Materials and Methods

### 2.1. Science Ethics and Informed Consent

Permission from the Vhembe District Municipality (VDM) was granted prior to visiting the study sites and conducting the study. Thereafter, an application was submitted to the Tshwane University of Technology (TUT) faculty committee for research ethics, and it was approved. Consent was sought from the tribal offices, which serve as the gatekeepers to clusters of villages, and this was granted with the condition that the headman or chief in each village be visited personally to give their own informed consent. Lastly, the purpose of the study was explained to each interviewee at household level before being asked for permission to partake in the study. Simple random sampling of villages and households was undertaken. Consequently, the study included only adult participants who were full-time residents of their respective households and possessed a basic understanding of at least one of the languages, namely, Tshivenda, Xitsonga, English, Sepedi or Setswana.

### 2.2. Site Description

The study was conducted in the VDM, which is located in the northern part of Limpopo Province, South Africa. It shares borders with the Southern African Economic Development Community (SADEC) and countries such as Mozambique, Zimbabwe and Botswana. It has an average annual rainfall of 600 mm, filling the perennial rivers in the wet season. The Luvuvhu catchment formed the central part of this study. It is the drainage point of all rivers in the VDM before emptying into the largest river, known as the Limpopo River, which flows into the Indian ocean via Mozambique. The average annual temperature in South Africa is 17.5 °C with the mean temperature in the warmer months (December and January) at 22 °C and 11 °C in the cooler season (June and July). The VDM is in the warmest parts of the country and maintains annual temperatures in the higher range of the scale at around 25 °C and higher in most months [29]. The most prevalent spoken languages are Tshivenda and Xitsonga.

The population is estimated to be 1,402,779 [30] and is made up of four local municipalities, as depicted in Figure 1. These are the Makhado, Collins Chabane, Thulamela and Musina local municipalities. A survey was conducted in three of these local municipalities. Musina Local Municipality was excluded from the study due to predominantly using groundwater as the main water source [8] and additionally being the least populated. The municipal profile analysis reports household users of boreholes making up approximately 9% of the district municipality. It was hence decided that the selected three local municipalities would represent the district. Thulamela is the most populated, at a head count of 497,237 [31].

### 2.3. Questionnaire Design

The questionnaire was designed in a semi-structured format, incorporating both closed-ended and open-ended questions, to accommodate diverse responses and provide valuable insights. In this manner, the gathering of data was accurate, yet flexible enough to give the respondents and the interviewer freedom to converse freely in order to assure security and confidence, and therefore source more honest answers.

Community engagement was carried out for a general understanding of the locals, as they are directly affected by the state of WASH and their immediate social economic context as depicted in Figure 2. The design was aimed at evaluating the objective of WASH strategies that are already in place being effective in waterborne disease prevention according to the current daily practices and circumstances.

### 2.4. Study Survey on WASH

The study was conducted using a cross-sectional design, and the sample size was determined to be 5% of the population in each village based on the regional statistics from the local municipality. A questionnaire was used to gather information determining the socio-demographic stature and WASH-related factors within the communities, potentially elevating risks to human health. Male and female respondents were not discriminated. Three DWTPs supplying the selected villages and the only WWTP linked to the water network were included in the study (Figure 1). The interviews were conducted from Monday to Friday in the daytime during the period of March 2020 to March 2021, while adhering to COVID-19 restrictions in place at the time. A total of 133 interviews were conducted with 51 residents in Ka-Mhinga, 40 in Ha-Mutsha and 42 in Maniini villages.

### 2.5. Water Sample Collection

A total of 360 water samples were collected (Table 1) from March to August 2021. In each of the three villages located within the respective local municipalities, a distinct distribution network was observed from the catchment to the DWTP and to the end user. Water was sampled at various points in the three cycles, both up- and downstream of the water sources supplying the DWTP, raw and finished water inside the DWTP and at the point of use. The only WWTP that was functional in the water network was also sampled. The DWTPs fed directly from natural surface water bodies which formed part of the study site. Water samples at the households were collected from the yard taps, most of which were located a few metres from the house. The taps were run for a few seconds before collection. Most houses stored water owing to the periodic water shortages. On the days that sample collection took place without running water, the participants would offer the stored water. The samples were treated the same.

### 2.6. Isolation of Bacteroides

This study ran concurrently with a study on protozoan parasites, where water samples were collected in 25 L polycan drums and filtered using Pall Envirochek^TM^ 1 μm HV filter capsule (Pall South Africa (Pty) Ltd., Midrand, South Africa). The supernatant solution (500 mL) used from protozoa recovery was filtered through a 0.2 µM mixed cellulose ester filter [32]. The bacteria trapped in the membrane were washed with Tween^®^ 20 and antifoam before being centrifuged at 1500× *g* for 20 min. The supernatant was discarded, and the pellet stored at −80 °C before being further processed. The DNA was extracted using Zymo Research Quick-DNA Fecal/Soil Microbe Miniprep Kit (Zymo Research, Inqaba Biotechnical Industries (Pty) Ltd., Pretoria, South Africa) following the manufacturer’s instructions. Briefly, ≤250 mg of the pellet was added to a ZR BashingBead™ Lysis with buffer and then centrifuged in a microcentrifuge for 1 min; this was followed by precipitation and purification using provided wash solutions, and then finally eluted with 50 µL of elution buffer.

### 2.7. Molecular Identification of Isolates

For *Bacteroides*, the extracted DNA was subjected to qPCR for the detection of faecal pollution sources, using host-specific (human, cow, pig, chicken and dog) marker assays. The primers and probes used for each marker are depicted in Table 2. The qPCR assay was performed with the total reaction volume of 20 µL, containing 10 µL of iQ™ Multiplex Powermix (Bio-Rad, Hercules, CA, USA), 2 µL of primer/probe mix, 6 µL of PCR-grade water and 2.0 µL of template DNA [33]. The assay was run in a Bio-Rad CFX96 Touch™ Real-Time PCR Detection System. The following cycling conditions applied: 95 °C for 30 s, followed by 45 cycles at 95 °C for 5 s and 60 °C for 30 s. No-template controls were used in every run, along with negative controls and the plasmid DNA containing the genetic markers specific to *Bacteroides* for the PCR assay.

The specificity, sensitivity and accuracy of the marker assays for the geographic region were tested in a separate and similar study in our laboratory. The DNA used was from faecal samples from the selected hosts [38]. The study considered the number of samples that were either true negatives or true positives and those that were either false negatives or false positives to formulate equations to calculate the sensitivity, specificity and accuracy.

Briefly, the specificity of *BacHum* marker for humans was 97.00%, the *BacCow* marker for cattle was 92.00%, *Cytb* for poultry was 95.00%, and lastly, the swine marker, *Pig-2-Bac* was 100.00%. The canine marker *BacCan* was 97.00% specific for dog faecal samples. The sensitivity was determined to be 100.00% for both human and cow markers, 71.00% for pig, 80.00% for chicken and 75.00% for the dog markers. The accuracy was also determined to be between 93.00% to 98.00% across the markers, with the highest values assigned to human and cow markers and the lowest to chicken and dog markers. The limit of detection for the primer sets were determined to fall within 26.17 and 31.65 gene copies per µL whereas the cut-off values ranged from 37.88 to 39.94 gene copies per µL for all markers.

### 2.8. Statistical Analysis

Results were entered into Stata/SE Version 14.1. software (StataCorp 2015). Bivariate linear regression analysis was used to represent the association between the presence or absence of markers of host-specific faecal contamination (binary outcome variable/y) in households against WASH practices (quantified predictive variable/x). Descriptive statistics were used to interpret the smaller observations of the catchment and treatment plants.

## 3. Results

### 3.1. Survey Findings

#### 3.1.1. Socio-Economic Status of the VDM

The respondents consisted of 39 males and 94 females. The oldest interviewee was 84, with the youngest being 22. The average age of the respondents was 45.70, with a standard deviation of 17.48. Households had on average 4.5 members in a family, with the largest family having 12 members. The highest number of children in a household was six (from 0 to 18 years). The households were largely made up of women and children. Of the 132 households, 71 had at least one person working, leaving 46.22% of the population without basic employment. The population showed a peak of young male adults and a steep decrease in females compared to males in the older age ranges (Figure 3).

#### 3.1.2. Water Systems

All three communities used unprotected municipal piped water to the yard. In addition, communal taps were present at designated street corners in Ka-Mhinga. Only groundwater sources, mostly as boreholes, in individual properties were protected. In all three communities, some respondents made use of untreated raw water from the river for daily duties, not limited to domestic use. Nearly 89.00% of the respondents stored water owing to the erratic supply (Table 3). From the population, 124 of the 133 households (approximately 93.23%) used piped municipal supply (improved drinking water sources) from unprotected water bodies (Figure 4). Only a few households were recorded as direct users of untreated water from the surface water sources.

#### 3.1.3. Sanitation, Hygiene and Health

The survey revealed that 99.25% of household members use sanitation facilities, with the remainder practising open defecation. A few (six) of the participants agreed to the usage of the facility by children under five years, yet disclosed safety concerns for children of less than five years of age to use the household sanitation facility. Those who answered “no” either have no children of that age (106) or have infants using diapers (21) (Figure 5).

Wastewater created within households was often disposed of in pit toilets or in the household backyard dumping site, where wastes would usually be buried or burnt (Table 4).

When asked about frequently occurring illnesses in the household, the majority of the replies (74.44%) were none. Following this, the incidence of diarrhoea was reported as 14.29%, then body rash, other diseases, and lastly, trachoma. Only 10 respondents disclosed that children less than five years old suffered from diarrhoea in their households. It was also discovered that many of those cases lasted 2 or 3 days with largely watery diarrhoea rather than mucopurulent stools. Stools with blood were not reported.

Among the age group of 50–65 years, diarrhoea was reported by 3.01% of respondents, and 3.76% acknowledged an incident with family members above 65 years. The last known occurrence of diarrhoea was reported as being mainly a week before. Members of the household presenting with diarrhoea are generally taken to the clinic and not anywhere else. All the respondents mentioned that no measures were taken to isolate the ill person until they recovered.

#### 3.1.4. Animals Found in Villages

Observations during the survey indicated the presence of wild birds and primates around water sources along with domesticated animals. The interviews gave more insight on animals within households (Figure 6).

#### 3.1.5. Water Treatment Plants

The treatment works that formed part of the study made use of a municipal laboratory for sample testing, which was centrally located at the VDM site. Potable water samples were sent there weekly or bi-weekly. It was also found that all the treatment plants had chlorine readily available all the time. The plants did not report any kind of vandalism. Some of the pumps and sludge treatment tanks at the Thohoyandou WWTP required repairs to function optimally (Table 5). The WWTP discharges its treated wastewater into the Mvudi River, which is a tributary of the Luvuvhu River, upstream of the Nandoni Dam. Maturation ponds were part of the treatment system on site, becoming more useful in events of mechanical breakdowns.

### 3.2. Sources of Faecal Pollution

The interviews revealed that three households in Ka-Mhinga had some animals when no others did. These households also cultivated land within their yards. Animal ownership was the lowest in Ha-Mutsha and the highest in Maniini. In terms of detecting host-specific faecal markers, cow faecal matter was not detected in Ha-Mutsha households, while pig faecal matter was detected there only once, followed by incremental increases in the detection of human and dog faecal matter, and lastly, the chicken-specific faecal marker was detected most frequently. Figure 7 illustrates the source of faecal contamination at the POU in the study sites as determined by qPCR.

Overall, in Ka-Mhinga households, the dog-associated marker at a detection rate of 33.33% was the most prevalent, while in Maniini, it was 18.42%. Moreover, Ka-Mhinga exhibited the highest poultry source of faecal contamination at 28.89%, followed by Ha-Mutsha at 15.00% and lastly Maniini at half of the contamination found in Ha-Mutsha. Ka-Mhinga shows a higher occurrence of faecal contamination and Ha-Mutsha the lowest.

The sources of faecal contamination at treatment plants—drinking water and wastewater—are presented in Figure 8. The Vondo WTP supplying Ha-Mutsha village presented the lowest faecal contamination, with none of the markers detected in the final water. A similar observation was noted in Nandoni WTP, supplying Maniini village, showing the human-associated marker in raw water only. A similar but higher detection rate was observed for Mhinga final water. The highest faecal contamination is recorded in the wastewater effluents as well as in Ka-Mhinga raw water (abstraction point). The human-associated marker was detected most frequently across the sites.

Overall, the surface water sources were found to be polluted with faecal matter, with all the target markers detected, except for the Vondo Dam (Table 6). The Nandoni Dam displayed all five faecal markers, with the pig-associated marker being the most prevalent at 87.50%. The Vondo Dam, on the other hand, presented a lower detection rate of the target markers; the human-associated marker was the most prevalent at 25.00%.

The regression analysis (Table 7) reflected overall varying relationships from no relationship (0.0) to moderate association (0.4). Out of 55 observations, where 11 WASH variables were each run against the 5 target markers, 14 (25.50%) presented a moderate relationship (R^2^ = 0.3–0.5) with a *p* value that was smaller than alpha (0.00–0.01). The highest of these was variable (e), water treatment in household (R^2^ = 0.414; *p* = 0.00), followed by (f), presence of wash basin (R^2^ = 0.413; *p* = 0.00), both associated with chicken faecal contamination. Weak relationships (R^2^ = 0.1–0.2) were observed for 30/55 (54.50%) observations. From those, only 20.00% were found not to be significant, with a *p* value that was greater than alpha (0.05). The remaining 80.00% were significant (*p* = 0.00–0.049).

## 4. Discussion

There remains a need to support and capacitate rural communities to monitor their daily drinking water quality and better manage the sanitation process [39]. Based on this premise, we sought to investigate the link between WASH practices and the presence of faecal contamination in drinking water supplies. The survey revealed that more women than men were found to be available in the households (Figure 3), thus confirming their role as primary caregivers, as described by WHO and other researchers [40,41]. This factor reiterates the need for focused interventions that will make a direct impact on the lives of those who carry much of the care and the domestic burden, giving women and children a voice in their communities. In this study population, more males were of a young age, pointing to the burden on frail elderly women for ensuring that there is sufficient water for domestic uses in households. The findings concur with the percentage of women (91.84%) fetching water far outnumbering the percentage of men (3.06%) (Table 3). The WHO/UNICEF [42] outlines the dire need to move away from the distinctive effects of water scarcity on gender roles, as women are often disproportionately affected by water scarcity. It seems a distant reality in this study population with such polar numbers of males or females fetching water. Unless serious intervention is effectively implemented, the female members of the community will continue to be negatively impacted by the lack of WASH.

The depiction of the gender ratio (Figure 3) illustrates a sharp peak of males in the age range of 22–31 years and a subsequent sharp drop in the 32–41-year age group. The authors Stecklove and Menashe-Oren [43] elaborate on gender ratios being driven by determinants such as rural–urban migration and employment. Males in their thirties migrate to find work, while their counterparts stay at home to provide care for the young. The authors elaborate on the trajectories that rural youth would follow, suggesting an increase in sub-Saharan African youth (15–24-year age group) up to the year 2050; this was also evident in our findings. This projection speaks to the large presence of under-five children there today, as well as the high fertility rate among young couples, and the emphasis to protect these vulnerable young from enteric pathogens.

When considering the cost of reagents (Table 4), a majority (40.60%) of the respondents alluded to purchasing at least one or two cleaning agents per month, spending over ZAR 10–49 (USD ±0.5 to 2). Following this were those spending ZAR 0 and 14.29% spending more than ZAR 100 (USD ±5) monthly. It was further found that close to all participants admitted to the capacity of purchasing cleaning agents and soap for washing hands. There was a weak association between the presence of markers in a household and the washing with soap practice, all having R^2^ values ranging between 0.089 and 0.257. The highest association in this range was the relationship with the chicken marker of faecal contamination, with a *p* value of 0.004 (Table 7). A study carried out on the knowledge and practices of caregivers on household water treatment in Ethiopia found that application of best practice was low even when the caregiver had knowledge of and a positive attitude towards safe drinking water in the household [44]. In our study, the knowledge of household water treatment methods was not investigated as a factor of educational level, because the information sought was not specifically from the primary caregiver. Although a larger proportion of the respondents were lacking higher education (figures withheld) it suggests little impact on the burden of faecal contamination at the household level.

The overall water coverage was high, finding 93.23% of respondents making use of improved drinking water sources abstracted from various natural water sources (Figure 4). Almost all the respondents (39/40) in Ha-Mutsha were satisfied with the supply or availability of water in their households (Table 3). This satisfaction could be attributed to the mixing of the municipal supply with personal boreholes. However, residents in villages like Maniini and Ka-Mhinga, relying more on municipal supply, expressed their grief about the availability of water in their communities. Additionally, there are worrying numbers of respondents storing water for everyday use (118/133). This not only applies to users of boreholes using untreated water but becomes riskier for users of household storage containers using utensils to draw water. The previously safe water loses integrity and becomes a breeding ground for potentially harmful pathogens [19,21,45,46]. To exacerbate matters, 87.30% of respondents do not practise any form of household water disinfection methods. Many respondents stored water for periods of a week and longer (Table 3). This ultimately leads to the need for water treatment at the household level, which in this case is barely practised. When water is treated and stored for prolonged periods, the risk of reinfection increases. Better sanitation in urban areas could be achieved by connecting households to sewer lines as well as through the basic hygiene practice of handwashing with soap [47]. It was established during the survey that even the water collection point itself, yard tap or communal tap, could be described as unsanitary, thus posing the irony of clean water from a dirty tap. Nonetheless, having the option of several water sources is an advantage for the villages.

Our findings reveal an applaudable 99.00% sanitation coverage. The use of pit latrines was the most popular at 63.00%, and open defaecation was the least popular at 1.00%. These impressive figures in a rural setting give hope for full coverage and show that the disregard for sanitation may be a thing of the past, as was found by Sibiya and Gumbo [17]. However, findings in Table 4 show 107/133 people without a handwash basin in the sanitation facility. A recent study revealed the risks at play after toilet use when hands are not immediately cleansed [48]. The authors also reveal the improvements to hygiene when a flush toilet is used. Exposure to faecal pathogens moves from latrines to fomites. It is, therefore, essential to immediately wash hands with soap after using the toilets.

The use of manure on farms necessitates good husbandry practices with proper veterinary care. This, in turn, would alleviate zoonosis [7]. Not only does bad manure management introduce pathogens into the water course, but Font-Palma [49] mentions the heavy metals, antibiotics, and the release of gases contributing towards greenhouse gases and salt toxicity. Above all is the critical need for water when farming. If the farmers were to use contaminated water without prior treatment for crop production, it may be a source of pathogens and the produce may be contaminated. The application of manure to agricultural fields must also be done with care; to disinfect manure for safer use requires resources that may not be readily available for local farmers. In our study, participants spend an average of ZAR 50 (USD > 2) monthly on cleaning agents. This is not nearly enough for maintenance of healthy disease-free livestock. However, in this regard, several methods described by Parihar et al. [50] may be applicable at affordable costs for local farmers. These are not limited to composting of faeces using handmade trenches, composting of animal carcasses and the various approaches to achieving good management of livestock wastes.

The results revealed that 76/133 respondents were not owners of animals. In addition, 55/133 observations found no animals around water sources (Figure 6). The most apparent indication of faecal contamination are cows, donkeys, goats and pigs around water sources. The mixed animal variable represents instances where the individual observations were made in combination with one or more of the other animals. The subsequent qPCR assays would determine that the dominant sources of faecal contamination were chickens and pigs in Nandoni Dam and the Luvuvhu River in Ka-Mhinga (Figure 9). Moreover, chickens, dogs and cats were not commonly observed around surface water sources but showed a higher presence at households in that order. Similarly, Cabral [51] showed dogs to have the highest faecal bacterial load compared to other domestic animals such as chickens, pigs and cats. This author found dogs to have 9.0 and 8.4 log_10_ cells/g of wet weight faeces of faecal streptococci and *Clostridium perfringens*, respectively. Among warm-blooded animals and humans, the highest count for faecal indicator bacteria was reported for human faeces, and the lowest counts for wild animals with faecal indicator bacteria ratios that are 10-fold lower than those of domestic animals. Many domesticated animals roam around the village drinking water from leaking taps, puddles and storage containers in the households if not provided with a drinking bowl or trough. Increasing urbanisation and the strive for a better life raise security concerns, therefore necessitating higher measures of protection against crime. The high presence of dog markers may stem from this reason. Dogs are also often used by farmers to assist in guarding livestock while grazing. Similarly, livestock that are kept passively and allowed to forage randomly often have little veterinary care. This allows them to be reservoirs for disease-causing zoonotic parasites and pose a risk to public health [7].

Surveying the water treatment plants revealed them to be in operational condition. Only one (Mhinga DWTP) of the three DWTPs was experiencing some challenges, especially in the wet season when the pipes would be blocked or broken after storms, with no repairs being carried out. The only WWTP in the study was found to have problems as well, with some infrastructure needing replacement or repairs. The staff confirmed their satisfaction in their positions and duties, as well as the ability to calculate the dosage for the disinfection agent in secondary treatment in the DWTP. Work carried out by Dunkin et al. [52] suggests the contributing factor of biofilm interactions with water microorganisms in distribution systems. Holding this as true would mean that the water retention time inside the pipes would consequentially be a factor as well. The longer the water is held in the system, the higher the chances of variability of microconsortia [53]. The negligible act of regularly opening the faucet, or not, in this regard would be significant. In addition, the volume of water used from a faucet as a function of time could contribute towards water quality. However small the change in the quality of water, it can be significant when the focus is on drinking water. Although the water pressure in this system was not evaluated, the risk of infiltration exists due to an old infrastructure and common leaks, exacerbated by an erratic supply.

In terms of the overall sources of contamination as seen in Figure 7, Ka-Mhinga was observed as the most contaminated, with four markers detected, and Ha-Mutsha as the least contaminated. Although no indicators of faecal contamination were detected in Ha-Mutsha and Nandoni WTP finished water, the raw water abstracted for the Nandoni DWTP (25.00%) showed possible human faecal contamination (Figure 8). The Mhinga DWTP finished water showed the presence of human faecal contamination at a detection rate of 28.60%, marking the highest contamination among the finished drinking water samples, suggestive of insufficient processing of the water in the DWTP before distribution. The Mhinga DWTP raw water was found to be dominated by the cow marker at a detection rate of 37.50%, followed by 25.00% for the human and chicken markers. We assigned the high detection rates to the many activities that are constantly taking place at the abstraction point, as it is easily accessible to humans and animals, as well as the short distance between the abstraction point and the small treatment plant.

The presence of host-specific faecal contamination is seen with the pig-associated marker being the lowest among surface water sources and peaking in the Nandoni Dam. The Nandoni Dam displays the highest faecal contamination in this group. This goes hand in hand with the abundant anthropogenic activities taking place in and around the dam at the easily accessible shallow ends lying at the feet of several villages. Natural water sources have proven of more importance to rural dwellers than communities in urban areas in the Venda area, especially because locals rely heavily on agriculture as a form of income generation [54]. However, it is the combination of rural livelihoods and urban developments that contributes to water source deterioration and consequentially has a negative impact on the rural communities compared to urban residents. Our study found that although the study areas had a municipal water supply connection to the yard or in their street, water supply was inconsistent and there was a need for either an alternate supply or for storage. Surveying revealed Ka-Mhinga to be more rural than the other two villages, with Ha-Mutsha being the more urbanised among them.

The WASH variables analysed showed few to moderate relationships with one or more MST markers (Table 7). The chicken marker had significant associations with all but three WASH variables. This means that the presence of chicken faecal contamination was significant across the household standpipes in relation to those variables, whether they kept chickens or not. However, frequent cases of diarrhoea, crops around water sources and total number of members in a household were not significant relative to the chicken marker. The WASH variable of different household animals wassignificant for the human, pig and chicken markers, meaning that the presence of these markers was of notable importance in households with or without animals, whereas cow and dog were not significant. The WASH variables duration of water storage, method of drawing water, water treatment in household and presence of wash basin were found to show a significant relationship with all five markers. This reflects some of the basic and daily practices in a household. Simply storing water for use during a scheduled water cut could negatively impact the water quality and put lives at risk of faecal pollution, right in the home.

The dog-associated marker was the most detected across all sites, followed by human and chicken, with pig being the least detected. Moreover, Penakalapati et al. [12] found that inconsistencies are common when linking animal presence in a household with the prevalence of diarrhoea amongst children in the household. In our study, frequent diarrhoea was significantly linked to the occurrence of human and pig markers, showing a weak relationship (Table 7). Illnesses often occurring in the household showed a moderate relation to cow, dog, human and chicken markers. Cow and dog markers were further found to have a weak association with crops grown around water sources. It has been reported that humans that live in close proximity to animals are at an increased risk of zoonotic infection [12]. Contrary to this, the high detection rate of markers indicated widespread faecal contamination, even in households without livestock or pets. Animal faecal wastes need to be managed properly to lower the chances of sickness. Roaming of stray animals needs a form of control as well. The Venda area generates money through farming and ecotourism and is the fastest growing in the province [54].

Waste stabilisation ponds as a control measure in WWTPs can pose risks in terms of performance. Ponds can be easily overloaded, especially if the design was poor to begin with, or short-circuited, leading to reduced retention times and the discharge of inadequately treated effluent with higher microbial loads. It was observed that the specified personal protective equipment was not used by staff, thereby increasing the chances of aerosol or dermal contact with wastewater potentially harbouring diarrhoeic pathogens, among other harmful agents [55]. Due to Thulamela being overpopulated, the WWTP was overloaded [31].

One DWTP and the WWTP have reported increased challenges during storm events. This leaves the end-user at the mercy of climate change [56]. Flood waters introduce a broad range of pathogens into natural water sources. Floods also cause interruptions in the daily operations and consequently increase the chances of more water supply interruptions for users. This only exaggerates the risk of waterborne diseases, as communities would turn to storage as one of their options. Alternatively, water could be purchased from neighbours, in which case the distance to access safe drinking water increases, thus regressing to unimproved water sources, as defined by the World Health Organization (WHO). Unfortunately, this was found to hold true in the study.

In our study, animal ownership was not a contributing factor to disease prevalence within households. It is possible to find a marker in a household where there are no animals, but the neighbouring households keep animals. It is not uncommon to find animals existing nomadically between households that are located close together or in a community. This could be a possible pathway for pathogens to spread, especially in settings where sanitation or hygiene are lacking. The overall MST profile matched what was expected after the general observations carried out during the survey phase of the project. These findings should advise existing water and sanitation safety strategies to strengthen community accountability and drive action towards high-quality public health. Drinking Water Safety Plans and Sanitation Safety Plans proposed by WHO and global projects such as the Global Water Pathogen Project encourage community members to have a sense of ownership for this precious resource and work together with authorities in the efforts to protect water bodies from pathogens [57,58].

## 5. Conclusions

The target hosts investigated as sources of faecal contamination in water sources were found with varying prevalence in the catchments and at the household level. The detection of dog faecal matter being the most prolific in household water samples goes hand in hand with the general understanding of human relationships with companion animals and their presence around homes for general safety or for guarding grazing livestock in agricultural settings. Similar to that are the levels of cow markers in surface waters, where most of the time is spent grazing. Above all, human faecal matter is known to pose greater risks to public health than that from animals. Our study detected high levels of the human-associated marker in all the water samples, indicating the possibility of pathogenic microorganisms that would be especially detrimental to those with weaker immune systems like babies under the age of five years and the elderly above 65 years. The main source of water in the villages was found to be piped municipal water to the yard tap, as well as privately owned boreholes. The absence of alternative water sources leaves users at the mercy of purchasing water from those with boreholes in the event of water supply interruptions. The presence of livestock other than those targeted here needs to be investigated as contributors to water pollution. The reason for keeping animals was not investigated. Perhaps if it had, there would be insights into reasons for keeping animals, such as for commercial purposes, or keeping dogs for security, cats for rodent control or simply keeping animals for food. The reason would allude to steps that are necessary for animal movement control and to whose responsibility it might be to ensure that the appropriate measures are taken.

Better management of animal wastes is imperative, as is safer handling of diapers, toys, utensils, and good quality hygiene practices after animal contact. We recommend fencing and bridging of water sources to limit animal access. Additionally, on-farm treatment of water for consumption by animals and better management including treatment of animal faecal deposits before use may ameliorate zoonosis. There was inconsistent agreement between the microbial source tracking results and household surveys, suggestive of widespread sanitation issues, even in households without domesticated animals. The heavy reliance of this study on interviews rather than on observations skewed the true outcome, in that participants were in some instances unable to recall, for example, the last diarrhoeal episode. In fact, the bias was observed quite often when participants became uneasy with some sensitive information. We recommend future studies that better manage this aspect. The data gathered here, which should be useful to environmentalists, water engineers and the likes of town planners, managers and economists, can be used to troubleshoot and then develop new tactics for addressing the issues at hand. It would be beneficial to make use of more than one marker for each assay to close the window of doubt on the absence of faecal pollution upon the undetectable presence of one marker. We recommend the use of microarray cards or the coupling of MST with chemical source tracking (CST) for faecal pollution of water sources.

## Figures and Tables

**Figure 1 pathogens-13-00016-f001:**
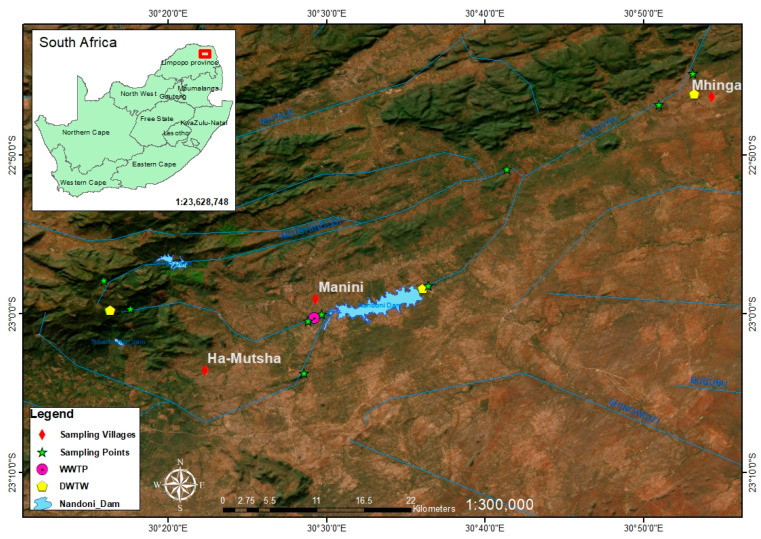
Map of study area showing the selected villages and the sampling points in the catchment.

**Figure 2 pathogens-13-00016-f002:**
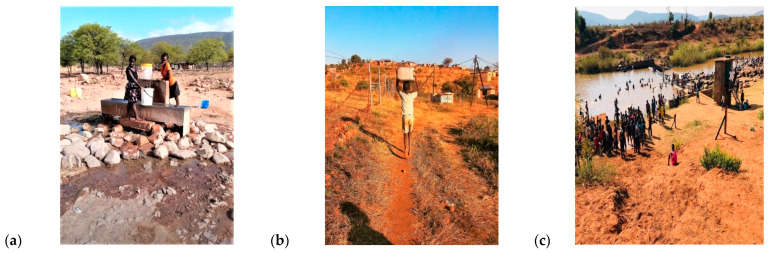
Photographs taken on site showing: (**a**) community drinking water tap; (**b**) a child walking after fetching water; and (**c**) various activities in a local river in the summer.

**Figure 3 pathogens-13-00016-f003:**
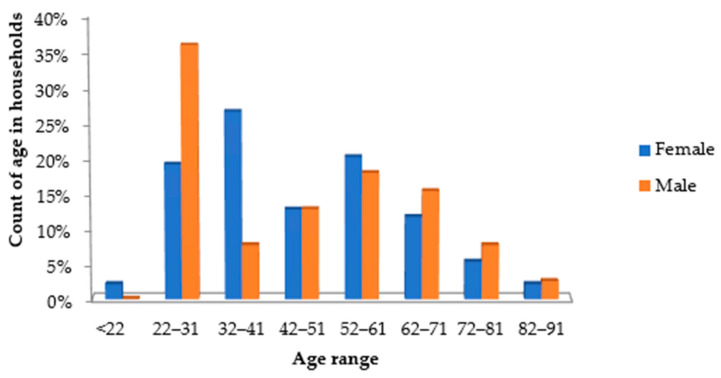
Demographic information of the study population.

**Figure 4 pathogens-13-00016-f004:**
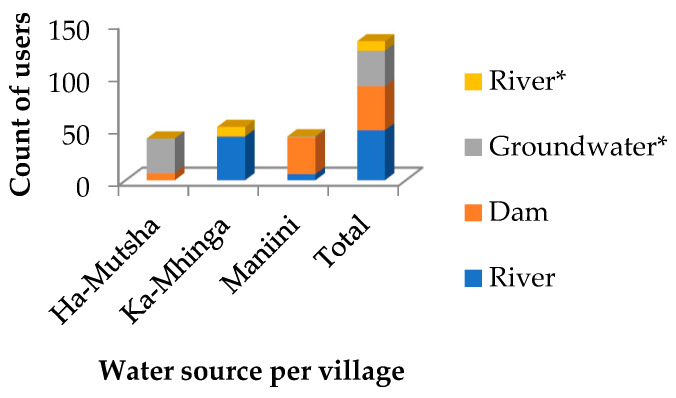
Main water source for the DWTP per village in the selected rural areas. * Direct use without passage through a DWTP.

**Figure 5 pathogens-13-00016-f005:**
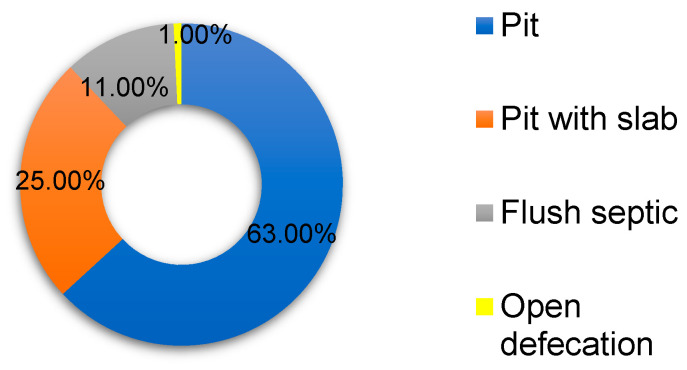
Sanitation facilities used in the study area (*n* = 133).

**Figure 6 pathogens-13-00016-f006:**
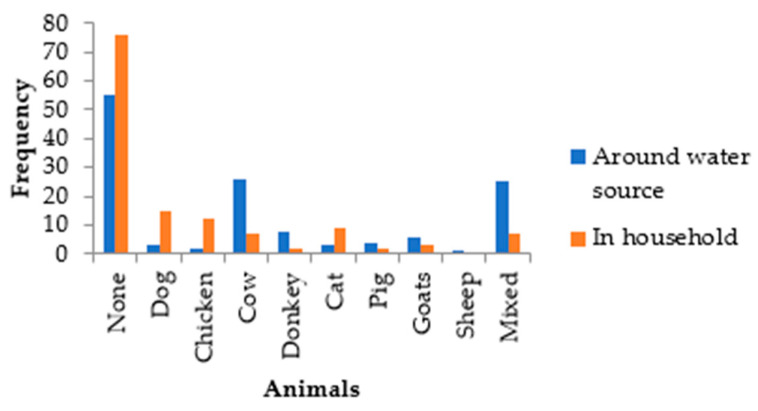
Animals around water sources compared to those found in households in the study area.

**Figure 7 pathogens-13-00016-f007:**
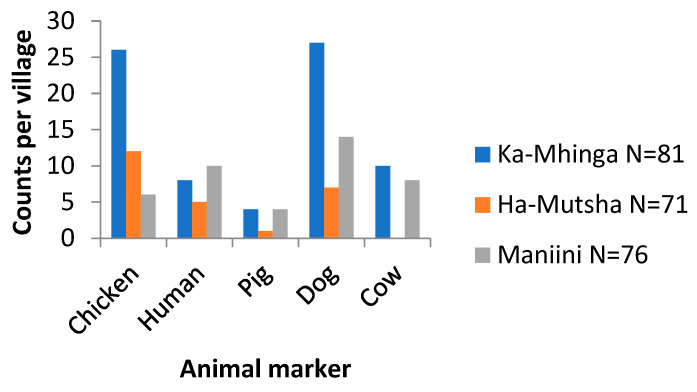
Overall sources of contamination per village in drinking water.

**Figure 8 pathogens-13-00016-f008:**
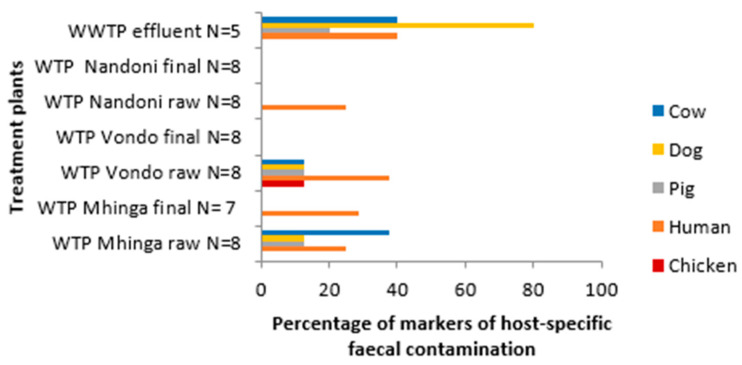
Markers of host-specific faecal contamination found among the tested water samples from the WWTP effluent and from each DWTP (raw water and finished water).

**Figure 9 pathogens-13-00016-f009:**
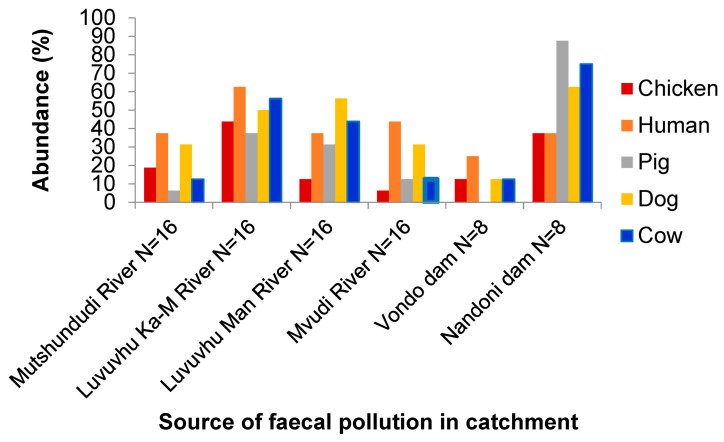
The presence of faecal contamination in unprotected surface water sources where abstraction of raw water for drinking water treatment occurs.

**Table 1 pathogens-13-00016-t001:** Layout of sample size for water collection.

Local Municipality	Village	Water Sources	Number of Households	Total Number of Samples
Makhado	Ha-Mutsha	5	10	111
Thulamela	Maniini	8	10	137
Collins Chabane	Ka-Mhinga	5	10	112
			Total	360

**Table 2 pathogens-13-00016-t002:** Oligonucleotides used for amplification of markers of host-specific faecal contamination (Anatech, SA).

Marker	Name of Primers and Probe	Primer or Probe Sequence 5′-3′	Reference
Human	BacHum160f BacHum241r BacHum193p	TGAGTTCACATGTCCGCATGA CGTTACCCCGCCTACTATCTAATG TCCGGTAGACGATGGGGATGCGTT	[34]
Cow	BacCow-CF128F BacCow-305r BacCow-257p	CCAACYTTCCCGWTACTC GGACCGTGTCTCAGTTCCAGTG TAGGGGTTCTGAGAGGAAGGTCCCCC	[34,35]
Pig	Pig2Bac41F Pig2Bac163Rm Pig2Bac113MGB	GCATGAATTTAGCTTGCTAAATTTGAT ACCTCATACGGTATTAATCCGC TCCACGGGATAGCC	[36]
Poultry	Chicken Cytb F Chicken Cytb R Chicken Cytb P	AAATCCCACCCCCTACTAAAAATAAT CAGATGAAGAAGAATGAGGCG ACAACTCCCTAATCGACCT	[37]
Dog	BacCan545f1 BacUni690r1 BacUni656p	GGAGCGCAGACGGGTTTT CAATCGGAGTTCTTCGTGATATCTA TGGTGTAGCGGTGAAA	[34]

**Table 3 pathogens-13-00016-t003:** Information gathered on water supply from the questionnaires.

Variable		Maniini	Ka-Mhinga	Ha-Mutsha	Total
Enough water *n* = 133	Yes	14	8	39	61
	No	28	43	1	72
Reason *n* = 72	Cut off	21	-	1	22
	Other	7	43	-	50
Male/female fetching water *n* = 98	Male	3	-	-	3
	Female	38	47	5	90
	Both	-	4	1	5
Agriculture around water sources *n* = 133	None	14	21	12	47
	Maize	14	6	5	25
	Tomato	2	3	1	6
	Beetroot	6	2	3	11
	Onion	6	2	3	11
	Spinach	-	1	4	5
	Other	-	9	6	15
	Mixed	-	7	6	13
Duration of water storage *n* = 133	Do not store	1	1	14	15
	1 day	5	5	-	6
	2 days	9	2	1	8
	3 days	4	14	1	24
	4 days	9	7	1	12
	5 days	12	11	-	20
	6 days and longer	2	11	23	48
Household water disinfection *n* = 133	Yes	1	5	1	7
	No	41	46	39	126
Household water disinfection methods known *n* = 133	None	13	25	12	49
	Bleach	11	8	7	26
	Boil	-	18	18	51
	Salt	16	-	-	2
	Other	2	-	3	5

(-) denotes no entry.

**Table 4 pathogens-13-00016-t004:** Sanitation and hygiene findings for the overall study area.

Variable		Response
Use of sanitation facility by children < 5 years	Yes	12
	No	121
Cost of cleaning agents *	Nothing	44
	Under ZAR 50	16
	Under ZAR 100	19
	Over ZAR 10	54
Toilet with basin	Yes	26
	No	107
Washing of hands	With soap	120
	No soap	9
	Sometimes	4
Disposal of household grey water	Veld	91
	Garden	2
	Building	1
	Other	39
Disposal of baby stools	None	115
	Dropped into toilet	3
	Disposed of outside premises	2
	Buried	1
	Disposed of into household waste	12
Frequently occurring illness	None	103
	Diarrhoea	19
	Trachoma	2
	Body lice	0
	Rash	6
	Other	3

*n* = 133. * Cleaning agents are any chemical, physical or mechanical means used for cleaning in homes.

**Table 5 pathogens-13-00016-t005:** Operations in DWTPs and WWTP serving the three villages.

Water Treatment Plant	Chlorine Used	State of Equipment	Plant Efficiency
DWTP 1 Nandoni	Gas chlorine	Good	Good
DWTP 2 Vondo	Gas and liquid calcium hypochlorite	Good	Good
DWTP 3 Mhinga	Gas and liquid calcium hypochlorite	Needs replacement	Pipeline often blocked or pump broken
WWTP Thohoyandou	Granular chlorine	Needs repair	Problems during floods

**Table 6 pathogens-13-00016-t006:** Origin of faecal contamination of water in the three villages on the basis of the host-specific faecal indicator markers found.

Sampling Site			Marker Occurrence in the Three Villages				
			Cow	Dog	Pig	Human	Chicken
River (*n* = 8)							
Mvudi	upstream WWTP		ND	2	ND	2	ND
	downstream WWTP		2	3	2	5	1
Luvuvhu	upstream Nandoni DWTP		ND	7	3	4	2
	downstream Nandoni DWTP		7	2	2	2	ND
Luvuvhu	upstream Ka-Mhinga DWTP		4	5	3	6	5
	downstream Ka-Mhinga DWTP		5	3	3	4	2
Mutshundudi	upstream Vondo DWTP		2	2	1	3	2
	downstream Vondo DWTP		ND	3	ND	3	1
Dam (*n* = 8)							
Nandoni			6	5	7	3	3
Vondo			1	1	ND	2	1
Water at the point of treatment (*n* = 8)							
Nandoni WTP		raw	ND	ND	ND	2	ND
		chlorinated	ND	ND	ND	ND	ND
Vondo WTP		raw	1	1	1	3	1
		chlorinated	ND	ND	ND	ND	ND
Mhinga WTP		raw	3	1	1	2	2
		chlorinated	ND	ND	ND	2	ND
Thohoyandou WWTP effluent			2	4	1	2	ND
Water at the point of use (*n* = 80)							
Maniini households supplied by Nandoni DWTP			8	14	4	10	6
Ha-Mutsha households supplied by Vondo DWTP			ND	7	1	5	12
Ka-Mhinga households supplied by the Mhinga DWTP			9	27	4	6	25
Communal tap in Ka-Mhinga			1	ND	ND	2	1

ND: Not detected.

**Table 7 pathogens-13-00016-t007:** Association of the presence of markers of host-specific faecal contamination detected in real-time PCR with WASH variables from the survey. (In bold are the higher r-squared values reflecting moderate relationships, having alpha less than 0.05).

MST Marker	Cow		Dog		Human		Pig		Chicken	
WASH Variables	R^2^	*p*	R^2^	*p*	R^2^	*p*	R^2^	*p*	R^2^	*p*
(a) Crops around water source	0.199	0.014	0.297	0.002	0.114	0.063	0.091	0.099	0.000	0.003
(b) Household animals	0.100	0.083	0.067	0.161	0.180	0.017	0.133	0.043	0.129	0.048
(c) Duration of water storage	0.195	0.013	**0.314**	0.001	0.233	0.006	0.157	0.027	**0.397**	0.000
(d) Method of drawing water	0.174	0.020	0.261	0.003	**0.313**	0.001	0.130	0.046	**0.398**	0.000
(e) Water treatment in household	0.147	0.033	0.272	0.003	0.245	0.005	0.155	0.029	**0.414**	0.000
(f) Presence of wash basin	**0.306**	0.001	**0.376**	0.000	**0.320**	0.001	0.133	0.043	**0.413**	0.000
(g) Hand washing	0.107	0.072	0.127	0.049	0.172	0.020	0.143	0.036	**0.310**	0.001
(h) Hand washing with soap	0.107	0.073	0.144	0.035	0.115	0.062	0.089	0.103	0.257	0.004
(i) Illnesses occurring in household	**0.367**	0.000	**0.435**	0.000	**0.345**	0.001	0.122	0.054	**0.421**	0.000
(j) Frequent diarrhoea	0.004	0.730	0.002	0.778	0.163	0.024	0.200	0.012	0.002	0.794
(k) Family total members	0.042	0.271	0.000	1.000	0.133	0.043	0.000	1.000	0.095	0.091

Questionnaire attached as Appendix A.

## Data Availability

All the relevant data are included in the article.

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
