# Peer review of "Assessing the Occurrence of Host-Specific Faecal Indicator Markers in Water Systems as a Function of Water, Sanitation and Hygiene Practices: A Case Study in Rural Communities of Vhembe District Municipality, South Africa"

_pathogens, 2023, doi:10.3390/pathogens13010016_

Round 1
Reviewer 1 Report
Comments and Suggestions for Authors
The manuscript is considered very interesting and complete, with important outputs.
Anyway, It would benefit the manuscript to give more details about the WWTP location ( in the map), and its description , in particular if the treatment system includes a final stage of maturation pond ( to disinfect the effluent) . It would also benefit the manuscript give additional details about the public water supply, in terms of risks of infiltration -water pipes are always under pressure or not, and if not, during how long in percentage?
A minor suggestion: Include the key-words by alphabetic order
Author Response
Thank you for the reviews.
The keywords have been ordered alphabetically.
The wastewater system has been further elaborated. This wastewater treatment works did have maturation ponds.
The treatment plants have been incorporated into the map.
The public supply water pressure is unknown however, the risk of infiltration exists due to old infrastructure and common leaks exacerbated by erratic supply.
Reviewer 2 Report
Comments and Suggestions for Authors
I found this manuscript interesting, the rational for the work clear and the topic of significance. The approach to the study also seems appropriate. However, there are two linked areas that weaken this manuscript.
1) Methodological detail
2) Statistics
Methodology
· Water sampling – if from taps where they run first then sampled?
· Sample volume – this was 25L for all samples?
· Samples for qPCR – were these used directly after extraction or were they stored prior to use?
· Can the authors give more detail on how the sensitivity and accuracy of the markers used was assessed – This is referenced but would like more information presented in the manuscript.
· A number of controls were run by the team for qPCR, however it would be important to know what is the limit of detection (cells) for each of the primer sets; a significant parameter linked to ND in table 6
· Figure 4 Main water source -I didn’t quite understand this figure is this the water sources used by the villagers? If so, this needs to be re-titled water sources per village.
Statistical analysis
· Simple linear regression would seem to be appropriate however, the data presented in table 7 is confusing.
· The layout is not easy to follow.
· The values highlighted reflect the coefficient values not the p values that relate to significance is that correct?
· The R2 values associated with some of this data suggest poor or no relationships between the two variables
· Does the fact that there are several coefficient values that at 0.44 mean that there are relationships between several of these variables?
I would suggest that this table and related data are revisited
Comments on the Quality of English Languageverall, the English is good but paragraphs like this (page 3 lines 105 onwards) could do with further proofing and reviewing.
Previous findings in South Africa reported the non-compliance of small-scale 105 wastewater treatment plants as well as potable water treatment plants in Limpopo Prov-106 ince as a matter of urgency with regards to public health. Not only did the compliance 107 with SANS 241 microbiological and chemical parameters fail, but so did the inability to 108 project data therefrom; which speaks volumes about the quality of operational perfor-109 mance of such plants. In 2016, Limpopo Province recorded the highest lack of safe and 110 reliable water supply [25]. A few years later, the numbers have improved, but much is 111 still to be done to meet the Sustainable Development Goals (SDGs) by 2030. Active and 112 competent authorities are essential for the success of the goal to reach full coverage of 113 water and sanitation in South Africa
Author Response
Methodology
Water sampling - the taps were run for a few seconds before collection.
Sample volume - all the samples were collected in volumes of 25 L and filtered, centrifuged for concentration and the supernatant from the centrifugation (500 mL) was passed through membrane filtration for Bacteroides isolation.
Samples for qPCR - DNA was stored prior to use at -80°C.
A more detailed assessment of the sensitivity and accuracy of the markers has now been included in the methods based on the cited reference.
The limit of detection and the cut off values have been indicated for the Not Detected samples.
Figure 4 - figure is titled main water source per village.
Statistical analysis
Table 7 - the layout has been revised.
The coefficient of regression has been removed for improved interpretation.
Quality of English language
This paragraph has been modified.